# Repeated dermal application of the common preservative methylisothiazolinone triggers local inflammation, T cell influx, and prolonged mast cell-dependent tactile sensitivity in mice

Jaclyn M. Kline[1], Erica Arriaga-Gomez[1], Tenzin Yangdon[1], Beebie Boo[1], Jasmine Landry[1], Marietta Saldías-Montivero[1], Nefeli Neamonitaki[1], Hanna Mengistu[1], Sayira Silverio[1], Hayley Zacheis[1], Dogukan Pasha[1], Tijana Martinov[2], Brian T. Fife[2], Devavani Chatterjea[1]*

1 Biology Department, Macalester College, Saint Paul, Minnesota, United States of America, 2 Department of Medicine, Center for Immunology, University of Minnesota Medical School, Minneapolis, Minnesota, United States of America

* chatterjead@macalester.edu

## Abstract

Occupational exposure to toxic chemicals increases the risk of developing localized provoked vulvodynia—a prevalent, yet poorly understood, chronic condition characterized by sensitivity to touch and pressure, and accumulation of mast cells in painful tissues. Here, we topically sensitized female ND4 Swiss mice to the common household and industrial preservative methylisothiazolinone (MI) and subsequently challenged them daily with MI or acetone and olive oil vehicle on the labiar skin. MI-challenged mice developed significant, persistent tactile sensitivity and long-lasting local accumulation of mast cells alongside early, transient increases in CD4$^+$ and CD8$^+$ T cells, eosinophils, neutrophils, and increases in pro-inflammatory cytokines. Therapeutic administration of imatinib, a c-Kit inhibitor known to inhibit mast cell survival, led to reduced mast cell accumulation and alleviated tactile genital pain. We provide the first pre-clinical evidence of dermal MI-induced mast-cell dependent pain and lay the groundwork for detailed understanding of these intersections between MI-driven immunomodulation and chronic pain.

## Introduction

Methylisothiazolinone or 2-Methyl-1,2-thiazol-3-one (MI) is a ubiquitous biocide preservative found in a wide range of household and commercial cleaning products, paints and hand and body soaps [1, 2]. It has come under scrutiny as a population-level risk factor for allergies and tissue injury in North America and Europe where 5–10% of populations assessed by patch-testing were found to be MI-sensitive [3]. MI exposure has been associated with tissue injuries in the skin [4, 5] and lungs [6–9]. Recently, Reed and colleagues have linked exposures to

**Data Availability Statement:** All relevant data are within the manuscript and its Supporting Information files.

**Funding:** This work was funded by National Institutes of Health 1R15AI113620-01A1 to D.C (https://www.nih.gov). JK, EAG, TY and SS were supported by summer stipends from an education grant from the Howard Hughes Medical Institute Foundation to Macalester College (https://www.hhmi.org). BB was supported by a Beckman Scholar Award to Macalester College from the Arnold and Mabel Beckman Foundation (https://www.beckman-foundation.org). JK, EAG, TY, BB, JL, MS-M, NN, HM, SS, HZ, DP were additionally supported by intramural summer research awards from Macalester College. The funders had no role in study design, data collection and analysis, decision to publish, or preparation of the manuscript.

**Competing interests:** The authors have declared that no competing interests exist.

household and workplace environmental toxins to an increased risk of developing vulvodynia [10]. Localized provoked vulvodynia is diagnosed by heightened sensitivity to pressure and touch and, in certain cases, accompanies increases in mast cells [11, 12], CD4+ T cells [13], and inflammatory cytokines in the vulvar tissue [14, 15]. It is relatively common, affecting an estimated 8% of individuals identified as women in studies of prevalence [16, 17]. The etiology of this chronic pain condition has yet to be fully elucidated and often, those experiencing this debilitating pain struggle to receive diagnosis and/or effective treatment, making further research necessary.

A history of seasonal and contact allergies doubles the risk of developing vulvodynia [18]. In pre-clinical rodent models of allergy-driven genital pain, multiple labiar or intravaginal exposures to common laboratory haptens—oxazolone (Ox) dinitrofluorobenzene (DNFB) provokes increased tactile sensitivity, tissue inflammation and local increases in mast cells [19–21]. We recently showed that repeated intravaginal MI (which is often added to vaginal washes) can cause sustained tactile genital pain accompanied by tissue mast cell increases in ND4 mice [3]. However, Reed and colleagues particularly identified *occupational* exposures (via salon work, housecleaning, yard maintenance etc.) as risk factors for developing vulvar pain [10]. In these jobs, and also other household hygiene practices, it is more likely that long-term repeated exposure to MI frequently, and most significantly, occurs via the skin. However, little is known about the inflammatory dynamics of MI contact allergy or whether such dermal MI exposures can also contribute to the development of persistent pain.

Here, we used our established murine model of provoked genital pain to measure changes in tactile ano-genital sensitivity, inflammatory cytokine profiles, and mast cell and T cell infiltration following MI application to the labiar skin. Repeated MI exposures triggered allergy-provoked pain in the labiar skin with considerably greater potency than in vaginal canal tissue. We also found that topical application of the small molecule tyrosine kinase inhibitor imatinib reduced mast cell abundance and pain sensitivity in MI-challenged mice. To our knowledge, these findings are the first evidence for the role of dermal MI exposures in inducing long-lasting, provoked pain.

## Materials & methods

### Animal usage

**Ethics statement.** 6-12-week-old naïve female ND4 Swiss mice (Harlan Laboratories, Indianapolis, IN, USA) were housed in Macalester College's animal facility, with a 12-hour light/dark cycle and free access to food and water. Mice were euthanized via 100% $CO_2$ inhalation at predetermined time points. This study was performed in accordance with the Guide for the Care and Use of Laboratory Animals of the National Institutes of Health. All experimental protocols were approved by Macalester College's Institutional Animal Care and Use Committee (IACUC B16Su2, approved on 1 July 2016; B19F2 approved on October 16, 2019). All efforts were made to minimize suffering.

For tactile sensitivity, we determined via our preliminary data on the effect of mechanical withdrawal latencies, 5% Type I error rate, and with the advice of our consulting biostatistician Dr. Vittorio Addona (Mathematics, Statistics and Computer Science, Macalester College) that 2 experimental repeats including at least 9 mice per treatment group would yield 80% power and enable us to detect relevant differences in our studies. For all other assays, using the same resources we determined that at least 3 mice per treatment group in 2 independent experiments would yield 80% power. Fewer than a total of 200 mice were used in these studies in accordance with these standards and in an effort to prevent unnecessary mouse usage.

**MI treatment.** MI sensitizations and challenges were adapted from previously described methods [3]. Mice were shaved on the back (15 mm x 15 mm), and sensitized the next day

(day 1) with a topical application of 100 μL of 1% MI (2-methyl-4-isothiazolin-3-one; Sigma Aldrich, St. Louis, MO) dissolved in a 4:1 mixture of acetone and olive oil (AOO; Sigma-Aldrich). On day 6, mice were re-sensitized with a topical application on the shaved back with 100 μL of 0.5% MI. On day 11, mice were shaved on the labia. Beginning on day 12, mice were subsequently challenged on the labia (5 mm x 5 mm) with 40 μL 0.5% MI or AOO control vehicle for 1, 3 or 10 days. Challenges were administered at the same time of day for the challenge period.

To collect skin for flow cytometric analysis, mice were sensitized with MI on the back as described above and shaved on the flank (25 mm x 25 mm) on day 11. On day 12, mice were challenged on the shaved flank with 25μL of 0.5% MI or vehicle per flank. Mice received 1, 3, or 10 daily challenges administered at the same time of day beginning on day 12.

**Imatinib treatment.**   We injected 100 μL of 30 mg/kg imatinib (Imatinib mesylate; Sigma Aldrich) dissolved in sodium hydroxide-buffered 0.9% saline intraperitoneally daily for 6 days beginning 1 day after 10[th] MI challenge and collected tissue 1 day after the 6[th] imatinib injection or 6 days following no treatment control (NT).

## Tissue collection and storage

Serum, spleen, flank tissue, labiar tissue, and iliac or inguinal draining lymph nodes were collected from mice euthanized by 100% $CO_2$ inhalation at various predetermined timepoints. Fat was removed from the flank and labia tissue samples. Tissue samples not used for flow cytometric analysis were flash-frozen in liquid nitrogen and stored at -80˚C. Tissues extracted for semi-quantitative real-time reverse transcription PCR (sqRT-PCR) were weighed prior to freezing and storage. To measure tissue eosinophil peroxidase (EPO) levels, tissues were weighed and stored in a weight-based volume of 0.5% hexadecyl trimethylammonium bromide (HTAB) buffer (Sigma-Aldrich) at -80˚C. Tissues used to evaluate myeloperoxidase (MPO) levels were placed in a solution of 50 mM $K_2HPO_4$ buffer (pH 6.0; Sigma-Aldrich) with 0.05% HTAB and stored at -80˚C.

## RNA isolation and quantification of gene expression

Total RNA was extracted from flash frozen labiar skin using the Total RNA Mini Kit (Midwest Scientific, St. Louis, MO, USA), eluted with RT-PCR grade water, quantified in ng/μL RNA using a Nanodrop ND-1000 Spectrophotometer (Thermo Fisher Scientific, Wilmington, DE, USA) or NanoPhotometer NP80 (Implen, West Lake Village, CA, USA), and reverse-transcribed in a 2720 Thermal Cycler (Thermo Fisher Scientific) using the Superscript III First-Strand Synthesis System (Thermo Fisher Scientific) using 100 ng of RNA per reaction. Relative transcript abundance was determined by semi-quantitative reverse transcriptase polymerase chain reaction (sqRT-PCR) using TaqMan Gene Expression Assay Primer/Probe Sets: *Interleukin-6* (*IL-6*; Mm00446190_m1), *chemokine C-X-C motif ligand* (*Cxcl2*; Mm00436450_m1), *Interleukin-1 beta* (*IL-1β*; Mm00434228_m1), *Interferon-γ* (*IFN-γ*; Mm01168134_m1), *T-box2 transcription factor* (*Tbx21*; Mm00450960_m1), and MasterMix (Life Technologies, Carlsbad, CA, USA) in a StepOnePlus Real-Time PCR System (Life Technologies). Results were normalized to expression of housekeeping gene *β-2-microglobulin* (*β2m*) and then calculated as fold-expression over vehicle controls [22].

## Protein quantification

Tissue levels of Cxcl2, IL-1β and IL-6 were evaluated as previously described [20, 21], from whole cell lysates using cytokine specific duo-set ELISA kits (R&D Systems, Minneapolis, MN, USA). Total serum Immunoglobulin E (IgE) content was measured by ELISA (Bethyl

Laboratories, Montogomery, TX, USA) in serum isolated from blood collected either from the sub-manibular vein or by post-mortem heart puncture. Absorbances were recorded with a PowerWave XZ microplate spectrophotometer (BioTek Instruments, Winooski, VT, USA) at 450 nm and 570 nm and the recorded optical density (OD) was used to determine protein concentration. Total protein concentrations in all samples were determined using a Detergent Compatible Assay (Bio-Rad, Hercules, CA, USA) following the manufacturer's instructions. Total protein concentrations were used to normalize target protein concentration found via ELISA.

## Quantification of local eosinophil peroxidase and myeloperoxidase

EPO and MPO analyses were performed as previously described [21]. For EPO measurements, labiar samples that had been stored for at least 24 hours in 0.5% HTAB buffer were thawed and 4x the original amount of HTAB buffer was added. The sample was homogenized, sonicated, freeze-thawed, re-sonicated, and centrifuged. Samples were then incubated with substrate solution, which contains 16mmol/L o-phenylenediamine (OPD; Sigma-Aldrich), 50mM Tris-HCl buffer, and 0.01% $H_2O_2$ for 30 minutes. Absorbance was measured at 490 nm. Tissues stored for MPO were thawed and homogenized in HTAB buffer and diluted 5-fold. The diluted homogenate was then sonicated, freeze-thawed, re-sonicated, and centrifuged. Absorbance was measured at 450 nm after a 20-min incubation in 50 mM phosphate buffer (pH 6.0; Sigma-Aldrich) with 0.025% $H_2O_2$ and 0.167 mg/mL o-dianisidine dihydrochloride (Sigma-Aldrich) at room temperature in the dark. Tissue EPO and MPO levels were normalized to tissue weight and optical density (OD) values per gram of wet tissue presented as OD/g.

## Flow cytometry

For inguinal lymph node analyses, flank-sensitized mice were sensitized on the back and challenged on the labia. For all other flow cytometric-based experiments, mice were sensitized and as described above. Draining lymph node (dLN) cells were isolated by crushing dLNs through a 70μm strainer. For skin cell isolation, leukocytes were isolated from flank skin samples as previously described [23]. Cells were washed, blocked with 1:100 anti-CD16/32 (Biolegend, San Diego, CA, USA), stained with fluorochrome-conjugated monoclonal antibodies (anti-mouse CD45.1, CD45.2, CD3, CD4, CD44, CD19; Thermo Fisher; CD3; BD Biosciences; CD19, CD25; Biolegend; CD8a, CD4, CD103, CD117 (c-Kit), FcεR1; Tonbo Biosciences, San Diego, CA, USA) at 1:100 dilutions for 30 minutes and analyzed on a BD LSR Fortessa X-20 flow cytometer (Becton Dickinson, Franklin Lakes, NJ, USA). Dead cells were stained with Ghost Dye-BV510 (Tonbo Biosciences, San Diego, CA, USA) and excluded from analyses. Data were analyzed using FlowJo Software (Version X, FlowJo LLC, Ashland, OR, USA; gating strategy is outlined in S1 Fig). Results were quantified as cell counts or mean ± SEM. Cell counts for each marker were calculated with their respective frequency of live cells and total cell count.

## Immunofluorescent staining and confocal imaging

Flash frozen labiar skin were embedded in Optimal Cutting Temperature compound (Sakura Finetek, Torrance, CA, USA) and cryosectioned into 12 μm sections. All sections were fixed in 4% paraformaldehyde (pH 8.5; Sigma-Aldrich). Sections for mast cell staining were permeabilized with 0.1% PBS Triton (PBST) (Sigma-Aldrich) and then blocked in 5% normal donkey serum/PBS (Jackson ImmunoResearch Laboratories, West Grove, PA, USA; #017-000-121). To stain mast cells, slides were incubated for one hour with Fluorescein-Avidin D (Vector Laboratories, Burlingame, CA, USA; #A-2001; 1:1000), as previously described [19, 24]. Following staining, all slides were cover-slipped with VECTASHIELD + DAPI (Vector Laboratories; H-1200).

Sections were imaged using a Zeiss LSM 800 laser scanning confocal microscope (Carl Zeiss AG, Oberkochen, Germany). Composite images of ten optical 1 μm sections projected on the z-axis were analyzed using ZEN 2.1 Imaging software (Carl Zeiss AG). Mast cell and nerve cell density was determined by fluorescent pixel intensity measurements, which were taken in four representative 5000 $μm^2$ regions of interest in each of three sections per slide for three slides per mouse. The average of four representative sections were taken, subtracted by a blank region of interest (empty slide with no fluorescence) to remove background, then divided by 5000 $μm^2$ to obtain the average fluorescent intensity/$μm^2$ for each section quantified. Values were then normalized to vehicle treatment group to obtain fold-change in average avidin fluorescent intensity. The experimental analyses were performed by experimenters blinded to experimental groups.

### Tactile sensitivity

Mechanical sensitivity was measured in a 2 mm x 2 mm area of the ano-genital ridge of mice with an electronic von Frey Anesthesiometer (IITC Corporation, Woodland Hills, CA, USA). Mice were six weeks old at the beginning of the experiment. Before taking sensitivity measurements, the mice were placed in individual Plexiglass von Frey chambers over a wire mesh grating for fifteen minutes to acclimate [19]. We have previously shown that hapten sensitization alone does not change tactile sensitivity neither does estrus stage [19, 20]. Two baseline measurements are taken at 1 and 2 days prior to sensitization for about twice the number of mice needed for the experiment. Only measurements between 0.1 and 2.5 grams were recorded— any measurement below or above the set range was re-measured. As previously described, mice with thresholds lower than 0.50 g and mice with the two baseline measurements differing by >1.00 g were excluded [19]. The mice with the highest baseline measurements were selected and then grouped into experimental and control groups so that the average within each of the two groups was as similar as possible.

Three to four experimental measurements were taken at different timepoints following ten challenges, by an experimenter blinded to the treatment groups. The average of these measurements per mouse was calculated for each timepoint and compared to baseline measurements as percent decrease from baseline. Based on published criteria [3, 19, 21, 25], mice were deemed to be hyperalgesic when they exhibited a 33% or greater reduction in experimental vulvar withdrawal thresholds compared to baseline. Any increase in sensitivity below the 33% decrease from baseline threshold is considered to be a negligible increase in tactile sensitivity.

### Statistical analysis

Data were processed using Excel (Microsoft, Redmond, WA, USA) or FlowJo Software (FlowJo, Ashland, OR) and graphed using PRISM 7 (GraphPad, San Diego, CA, USA). One-way ANOVA, *post hoc* Tukey HSD analyses, or unpaired Student's t-test were performed using PRISM 7 to compare treatment groups at designated time points. Statistical significances (indicated by asterisks) of $p < 0.05$, $p < 0.01$, and $p < 0.001$ were reported for differences between treatment groups.

## Results

### Repeated MI challenge induces increases in pro-inflammatory cytokines, and eosinophil and neutrophil activity in the labiar skin

We sensitized ND4 Swiss female mice with 1% and 0.5% MI dissolved in 4:1 acetone:olive oil (AOO) on the back and followed with 10 daily challenges of 0.5% MI or vehicle (AOO) on the

labiar skin. We assessed changes in expression of pro-inflammatory cytokines known to be elevated in vulvar biopsies [14] at the MI challenge site. 1 day after 10 challenges, mRNA transcripts encoding *Cxcl2*, *IL-1β* and *IL-6* were variably upregulated. *Cxcl2* and *IL-1β* expression showed a 1- to 6-fold increase and *IL-6* expression was elevated by 1- to 15-fold (Fig 1A). Increases in *IL-6* transcripts persisted 21 days after the 10th MI challenge, while elevated expression of *Cxcl2* and *IL-1β* resolved to baseline levels by that time (Fig 1B). Cxcl2, IL-1, and IL-6 protein concentrations were also significantly increased by 2- to 5-fold in the labiar tissues of MI-treated mice compared to their AOO-treated counterparts at 1 day after the 10th challenge (Fig 1C–1E). By 21 days after cessation of MI challenges, pro-inflammatory cytokine concentrations in the MI-challenged mice were indistinguishable from those treated with AOO (Fig 1C–1E). Given the increases in pro-inflammatory cytokines, we examined immune cell infiltration in the labiar tissue. Tissue eosinophil peroxidase (EPO) levels, a proxy for activated eosinophils, were increased 3-fold or higher in 10x MI-challenged skin than in AOO-treated skin (Fig 1F); these differences were resolved by 21 days (Fig 1F). In addition to increased eosinophil activity, we also found significantly increased tissue myeloperoxidase (MPO) levels, a marker for heightened neutrophil activity, in the labiar skin of MI-treated mice compared to AOO-treated mice 1 day after 10th challenge (Fig 1G).

## CD4+ and CD8+ T cell and mRNA transcripts associated with T cell differentiation increase in the labiar skin of ND4 female mice after 10 MI challenges

We evaluated T cell infiltration in MI-challenged skin using flow cytometry to further assess the inflammatory dynamics of the dermal MI response. 1 day after the 10th MI flank challenge in female ND4 Swiss mice, CD4+ cell numbers were significantly increased in the skin from 50 CD4+ T cells in AOO-treated to 300 in MI-treated skin (Fig 2A). Additionally, activation states of these cells expressing CD25 and/or CD44 were significantly increased in the MI-treated skin when compared to vehicle treated counterpart, with more than half of the CD4+ cells expressing CD44 (Fig 2A). With the exception of the CD4+CD44+CD25+ cells, T cells in MI-challenged skin remained elevated compared to vehicle treated skin at 21 days after 10th MI challenge, but this was not statistically significant (Fig 2A). For a more granular look at the earlier infiltration dynamics of these cells in the skin, we used 1 and 3 daily MI challenges along with the same sensitization protocol as our 10-challenge model. One day after a single challenge, there was no difference between CD4+ T cell populations in MI- versus vehicle-treated mice (Fig 2B). However, one day after three challenges, we observed a significant increase to almost 600 CD4+ T cells in MI-treated skin compared to 275 cells in vehicle-treated skin (Fig 2B). Activated CD4+CD44+ were also significantly increased with 300 of these cells detected in MI challenged skin and 150 in AOO challenged (Fig 2B). CD4+CD44+CD25+ cells were also increased at 1 day after 3 MI challenges, but, again, this increase was not significant (Fig 2B).

In addition to elevated CD4+ T cell population densities, CD8+ T cells were also increased (Fig 2C). At 1 day after 10th MI challenge not only were CD8+ T cells significantly elevated, numbers of these cells expressing CD44 or CD103 were also increased (Fig 2C). We observed 35 and 15 CD8+ cells in MI- and AOO-treated skin, respectively (Fig 2C). At 21 days after 10 challenges, levels of CD8+ cells in MI treated skin remained elevated when compared to vehicle counterparts (Fig 2C) and importantly, CD8+CD44+ and CD8+CD103+ cell numbers were significantly increased in MI-challenged skin with no detectable cells expressing these markers in vehicle treated tissue. Dynamics of CD8+ T cell influx were similar to those of CD4+ cells, where at 1 day after 1 MI challenge we observed little to no difference in infiltration of CD8+ cells and at 1 day after 3 challenges numbers of CD8+ and CD8+CD44+ were significantly

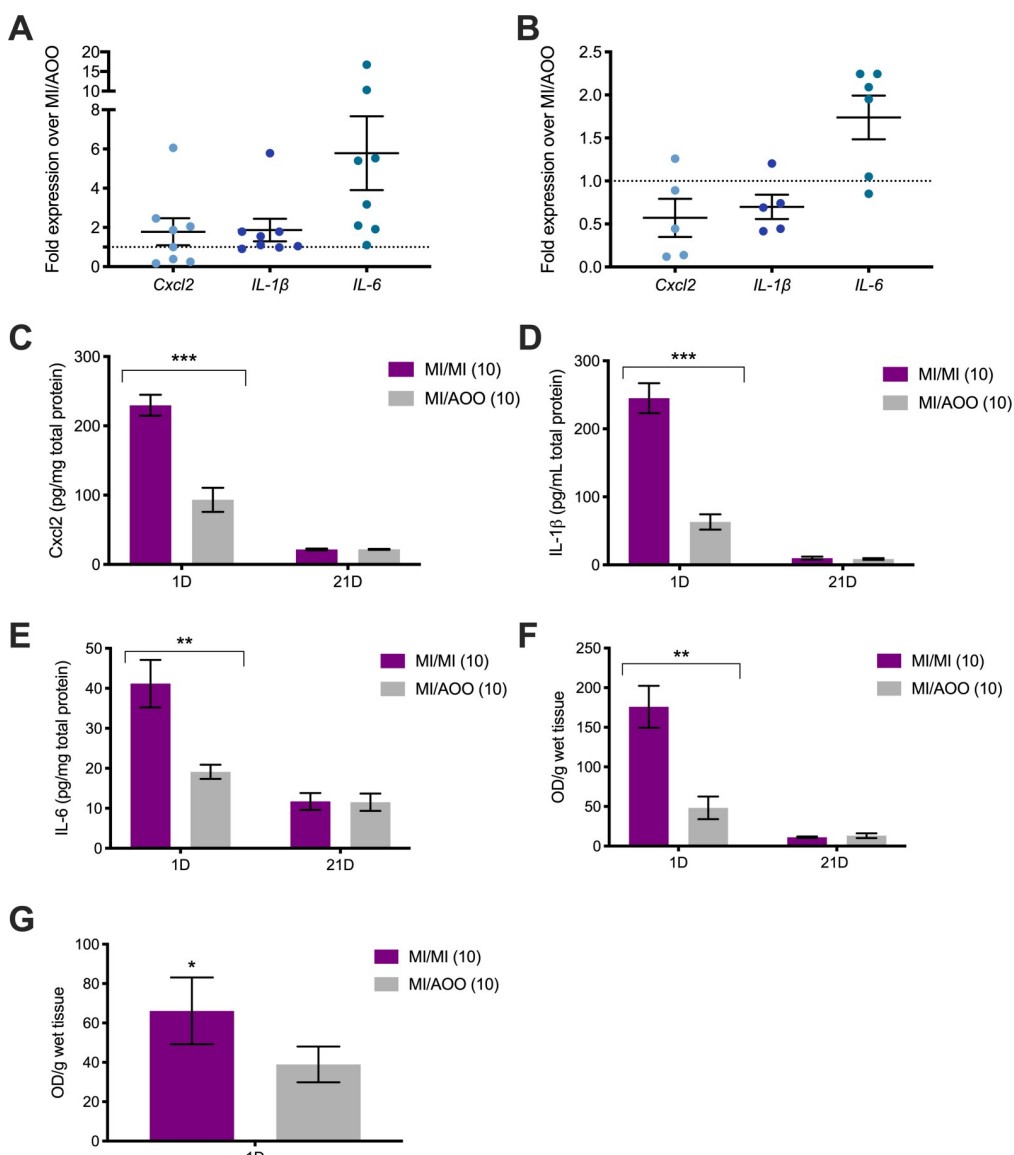

**Fig 1. Repeated MI challenge induces upregulated gene expression and protein content of pro-inflammatory cytokines, and increased eosinophil and neutrophil activity in the labiar skin of outbred ND4 Swiss mice.** (A) Transcript abundance of *Cxcl2*, *IL-1β*, and *IL-6* in labiar tissue extracted 1 day after 10 daily challenges on the labia relative to vehicle challenged controls. The black dotted line denotes y = 1. Results normalized to housekeeping gene *β2m* mRNA levels; n = 9/treatment group, representative of 3 separate experiments. (B) Relative transcript abundance of *Cxcl2*, *IL-1β, and IL-6* 21 days after 10 daily MI challenges. Black dotted line denotes y = 1. Results normalized to housekeeping gene *β2m* mRNA levels and vehicle challenged controls; n = 6/treatment group, representative of 2 independent experiments. (C-E) Protein concentration of Cxcl2 (C), IL-1β (D) and IL-6 (E) 1 and 21 days after 10 MI labiar challenges, determined via ELISA and normalized to total protein concentration from DC assay; n = 3-8/treatment group. Significance with respect to control group ** $p<0.01$, *** $p<0.001$. (F) Tissue eosinophil peroxidase levels measured by optical density/g wet labiar tissue 1 day and 21 days after 10[th] MI challenge; n = 2-7/treatment group, representative of 2 separate experiments. Significance with respect to control group ** $p<0.01$. (G) Tissue myeloperoxidase levels measured as optical density/g wet labiar tissue at 1 day after 10 MI challenges; n = 5-7/treatment group, representative of 2 separate experiments. Significance with respect to control group * $p<0.05$.

increased in MI-treated skin, with more than double the number of these cells compared to controls (Fig 2D). Interestingly, at 1 day after 1 or 3 challenges, CD8[+]CD103[+] cell numbers were not significantly different, although they were higher in MI-challenged skin than in their

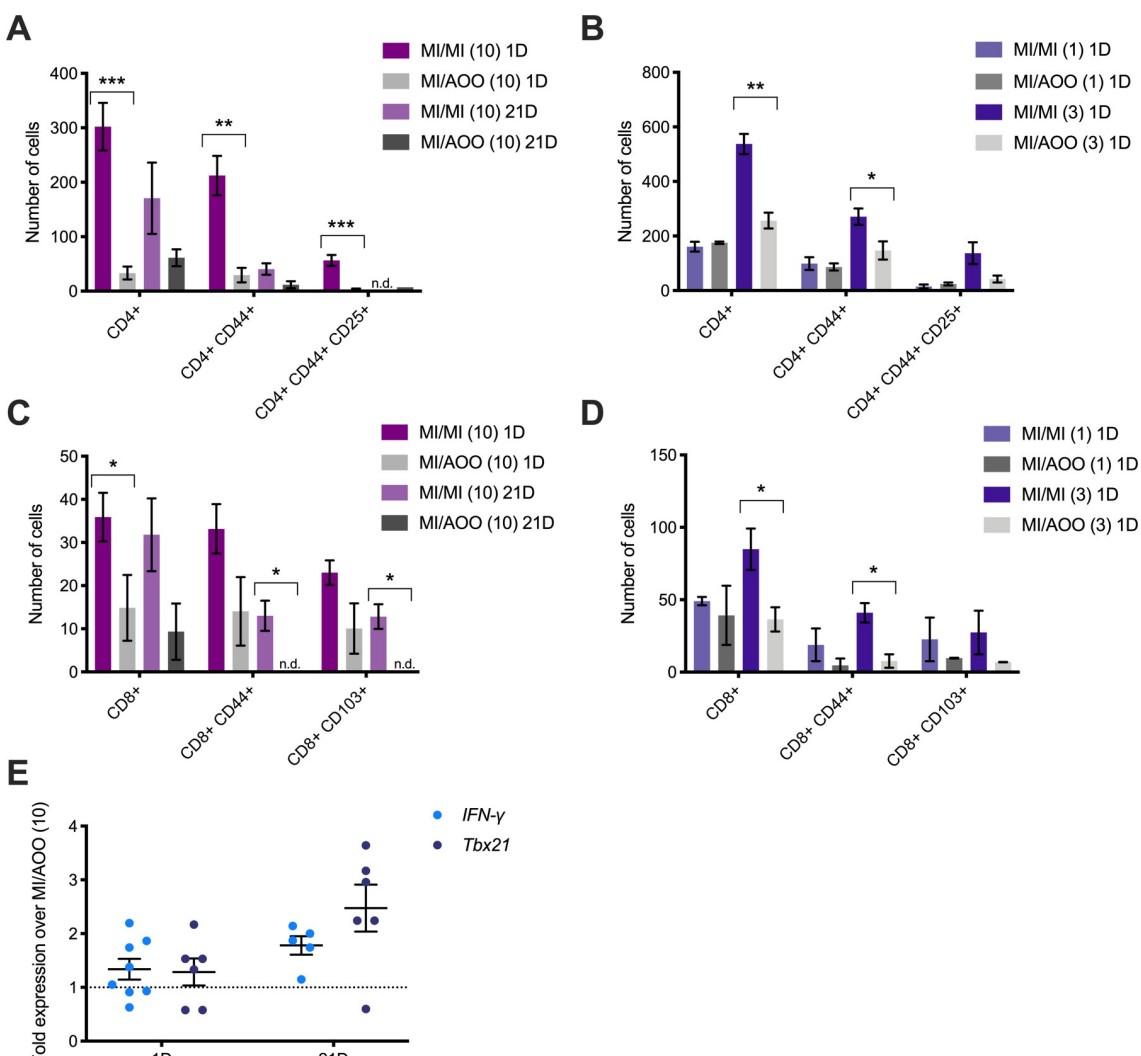

**Fig 2. CD4⁺ and CD8⁺ T cell infiltration, alongside upregulation of *IFN-γ* and *Tbx21* mRNA transcripts, is induced by repeated MI challenge in previously sensitized mice.** (A-B) Flow cytometric analysis of collagenase-digested CD4⁺ cells, gradient separated from MI- or AOO-treated flank skin collected at 1 and 21 days after the cessation of 10 challenges (A) or 1 day after 1st or 3rd challenge (B). (C-D) Flow cytometric based detection of CD8⁺ cells in flank skin collected 1 day or 21 days after 10th challenge (C) or 1 day after 1st or 3rd challenge (D). Total counts were calculated using beads and FlowJo gating counts and percentages, cell counts for each marker were calculated with their respective frequency of live cells and total cell count. Results displayed as mean ± SEM, outliers excluded via Grubbs testing; n = 3-6/treatment group, 1 sample represents 2 mice pooled representing 2 independent experiments. Results displayed as mean ± SEM, outliers excluded via Grubbs testing; n = 3-6/treatment group, 1 sample representing 2 mice pooled. Results with no detectable cells displayed as "n.d." Significance with respect to control groups * p<0.5, ** p<0.01, *** p<0.001. (E) Relative transcript abundance of *IFN-γ* and *Tbx21* in the labiar tissue at 1 and 21 days after 10th MI challenge. Results normalized to *β2m* mRNA levels and relative to vehicle-treated controls; n = 5–9 mice/treatment group, representing 2–3 independent experiments. Black dotted line denotes no change in relative abundance of transcripts.

AOO counterparts (Fig 2D). We evaluated levels of mRNA transcripts encoding *IFN-γ* in MI 10 challenged skin and found that at 1 day after 10th challenge levels were increased by an average of 1.3-fold; at 21 days there was a 1.8-fold average increase in MI-treated compared to AOO-treated mice (Fig 2E). Additionally, transcripts encoding *Tbx21*, an important driver of IFN-γ production [26], were increased by an average of 1.3- and 2.5-fold at 1 and 21 days after 10th challenge, respectively (Fig 2E).

## Repeated topical MI exposure increases mast cell density in allergic skin and draining lymph nodes, and elevates serum IgE concentration

We assessed changes in mast cell density in the skin of MI-challenged mice up to 63 days after 10th MI challenge. One day after 10 MI labiar challenges, we observed an increase in mast cells, which persisted up to 63 days after 10th challenge (Fig 3A–3E). The increase in mast cell density was significant up to 49 days post 10 challenges, with the exception of the 21-day timepoint (Fig 3A–3E). In addition to increased mast cell density, serum IgE levels were significantly increased in MI-treated mice 1 day after 10th challenge by almost 3 times the IgE concentration of vehicle treated counterparts (Fig 3F). This increase persisted up to 63 days after cessation of 10 challenges but was no longer significant. In addition to increased density of mast cells in the skin, 10 MI challenges induced significant early expansion of c-Kit+ mast cells in the inguinal draining lymph nodes early on. 1 day after 10th MI challenge there were double the number

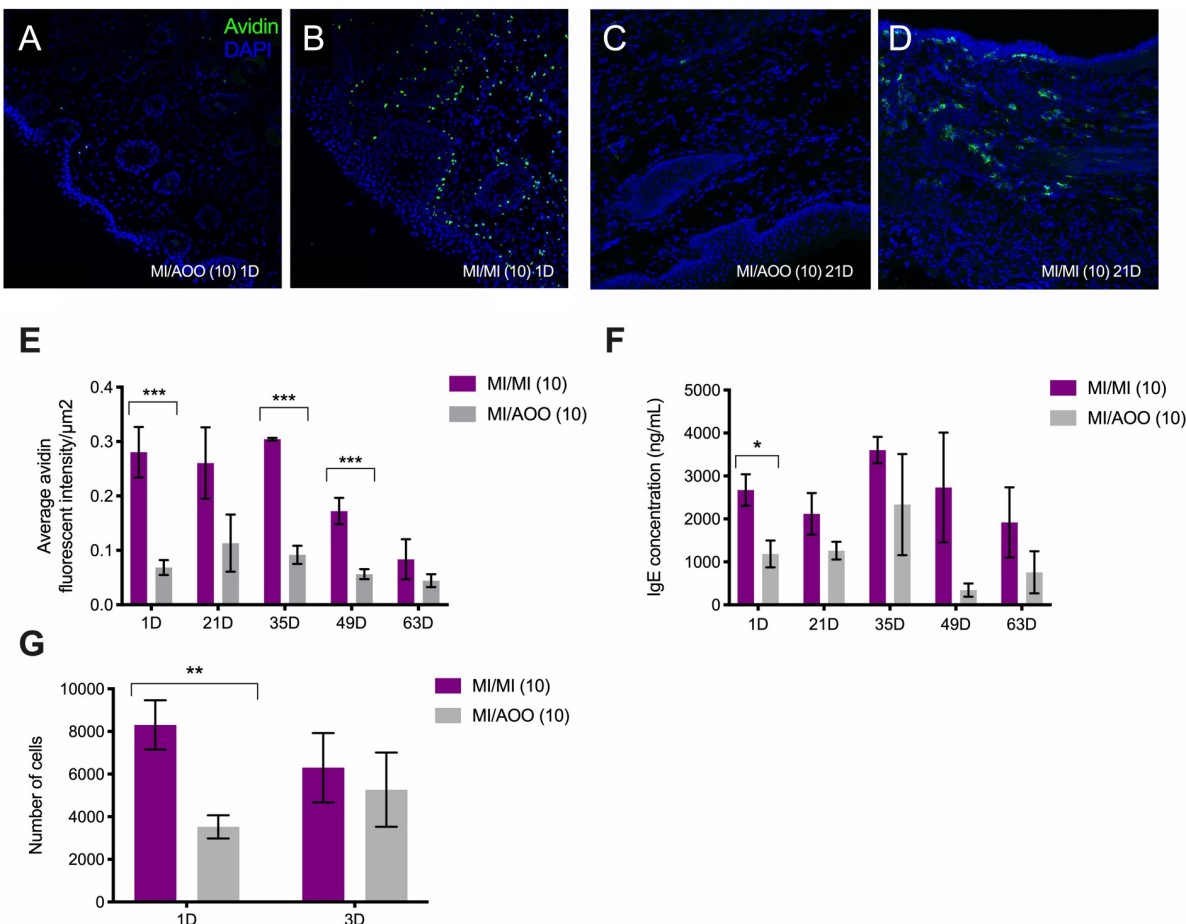

**Fig 3. Increased local mast cell density and serum IgE, and expansion of mast cells in the draining lymph nodes after 10 MI labiar challenges in previously sensitized ND4 female mice.** Representative confocal images of labiar tissue collected at 1 (A-B) and 21 (C-D) days from mice challenged 10 times daily with AOO (A, C) or MI (B, D) shown at 200× magnification. DAPI is in blue and avidin in green. (E) Density of avidin+ mast cells in 10 μm labiar skin cryo-sections from sensitized mice challenged with MI or AOO. Results reported as fold change in avidin signal in MI- over AOO-treated controls, displayed as mean ± SEM; n = 3-9/treatment group, representing 2 separate experiments. Dotted line denotes no change in signal intensity. (F) Serum IgE content in MI- or AOO-challenged mice at 1, 21, 35, 49 and 63 days after the cessation of ten daily challenges; n = 2-12/treatment group, representing 2 independent experiments. Significance with respect to vehicle control group* p<0.05. (G) Flow cytometric based detected of c-Kit+ mast cells in the inguinal draining lymph nodes 1 and 3 days after 10 daily flank challenges; n = 3-14/treatment group, representing 2 independent experiments. Significance with respect to control group, ** p<0.01.

of mast cells as compared to AOO challenged mice—8000 mast cells as compared to 4000 in AOO challenged mice (Fig 3G). At 3 days after 10 challenges this increase was no longer significant.

### 10 MI challenges induce a significant and persistent increase in tactile ano-genital sensitivity that lasts up to 70 days after challenge cessation

In concert with the inflammatory response, we found that previously sensitized female ND4 Swiss mice repeatedly challenged on the labiar skin with MI exhibited increased sensitivity to touch, as evaluated using an electronic von Frey anesthesiometer (Fig 4). This increase in sensitivity was significant and persisted up to 70 days after the 10th MI challenge (Fig 4, S1 Table). The level of heightened pain responses remained relatively constant through this period with MI-challenged mice showing withdrawal responses to ~50–60% less force than at baseline. The pain responses also lasted much longer than the early spike in inflammatory cytokines and infiltration of regulatory and memory T cells into MI-exposed labiar tissue (Figs 1 and 2). Notably, pain sensitivity appeared to be concomitant with mast cell increases in the labiar tissue that persisted (Fig 3).

### Therapeutic imatinib treatment significantly reduces mast cell density and tactile ano-genital sensitivity in ND4 female mice previously exposed to repeated MI challenge

The c-Kit-blocking therapeutic imatinib is known to reduce mast cell numbers in mice [27]. We administered imatinib daily via intraperitoneal injection for 6 days beginning 1 day after the 10th MI labiar challenge in previously sensitized ND4 female mice. 1 day after the 6th therapeutic imatinib treatment, mast cell density was significantly decreased in the labiar tissue of imatinib-treated mice compared to untreated MI-challenged mice (Fig 5A–5C). Alongside the decrease in mast cell density, we also observed a significant decrease in tactile sensitivity in imatinib -treated vs. control MI-challenged mice (Fig 5D). While MI-challenged mice

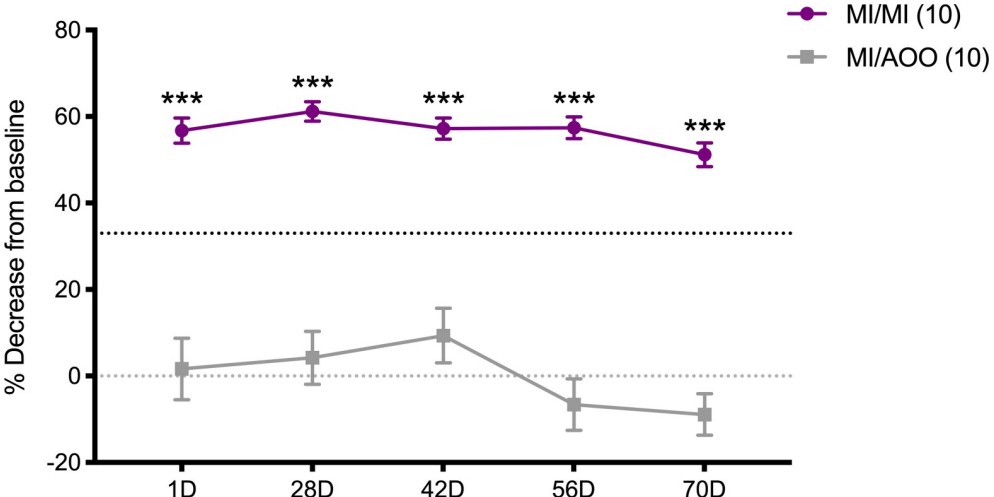

**Fig 4. Tactile ano-genital sensitivity significantly increased at various time points beginning 1 day after 10 MI challenges.** Tactile sensitivity in mice challenged daily for 10 days with MI or AOO, reported as mean ± SEM of the percent decrease from baseline in the withdrawal threshold for each treatment group; n = 41/treatment group for 1D timepoint otherwise n = 27/treatment group. Results represent 3 separate experiments for 1D timepoint, otherwise 2 separate experiments. Black dotted line denotes 33% decrease from baseline. Significance with respect to control group *** p<0.001.

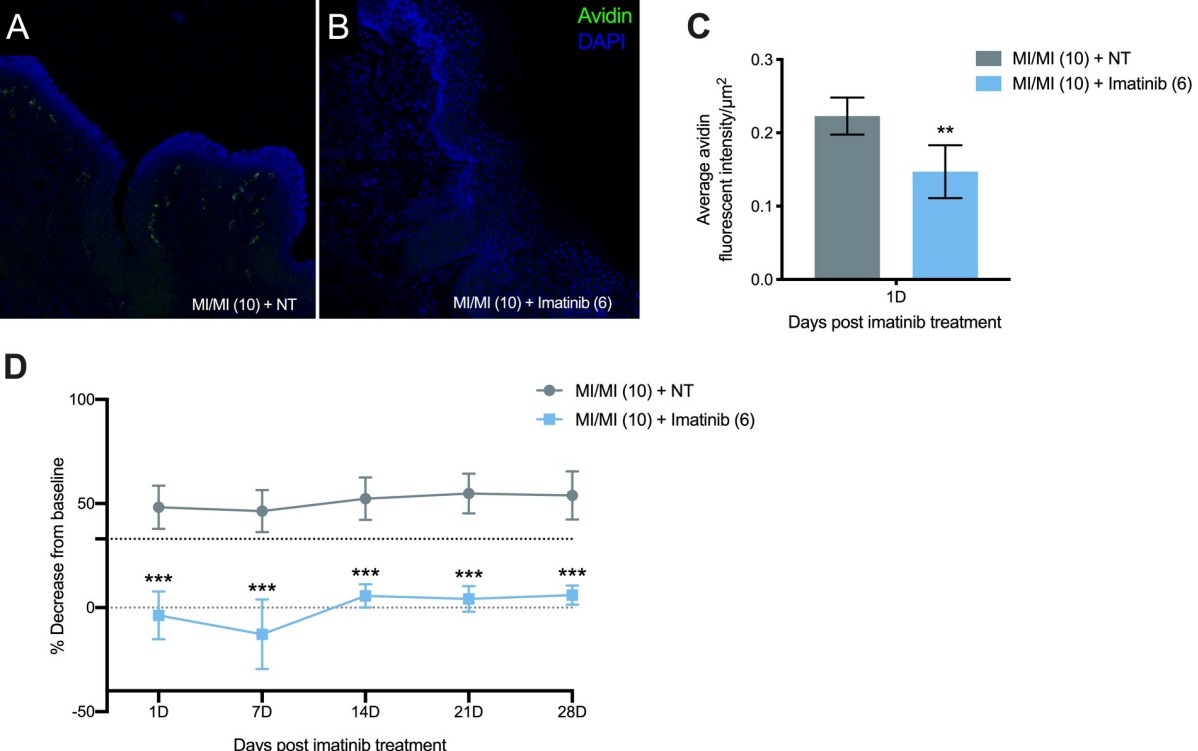

**Fig 5. Mast cell density and ano-genital sensitivity following therapeutic imatinib treatment in ND4 female mice previously exposed to MI.** (A-B) Representative images of labiar tissue collected from MI-challenged mice 1 day after 6 daily therapeutic imatinib treatments or NT shown at 200× magnification. DAPI is in blue and avidin is in green. (C) Quantification of avidin+ mast cell density in 10 µm labiar skin cryo-sections collected 1 day after 6 daily therapeutic Imatinib treatments or NT following 10 daily MI treatments in previously sensitized mice. Results displayed as mean ± SEM, n = 5-6/treatment group, representing 2 experiments. Significance with respect to control group, ** $p < 0.01$. (D) Tactile sensitivity in mice MI-challenged daily for 10 days and treated with therapeutic imatinib for 6 days. Results reported as mean ± SEM of the percent decrease from baseline in the withdrawal threshold for each treatment group, n = 8-9/treatment group, representing 2 independent experiments. Black dotted line denotes 33% decrease and grey dotted line denotes 0% decrease from baseline sensitivity. Significance with respect to control group, *** $p < 0.001$.

exhibited a 50% decrease from baseline sensitivity, mice treated with MI and imatinib showed almost no decrease from baseline (Fig 5D). This decreased sensitivity was sustained for up to 28 days after the 6th imatinib treatment, while untreated mice displayed elevated sensitivity that did not resolve by that timepoint (Fig 5D).

## Discussion

Vulvodynia is a poorly understood pain condition with no effective treatment. Alongside seasonal and contact allergies [18], dermal exposures to specific insecticides, herbicides, cosmetics, cleaners and solvents are also associated with a risk of developing vulvodynia [10]. Many of these products contain the biocide preservative MI. We previously reported that vaginal exposure to MI provokes short-term provoked genital pain in ND4 mice lasting up to 21 days after 10 intravaginal exposures to 0.5% MI in sensitized mice [3].

Here we investigated the inflammatory and nociceptive effects of dermal applications of MI which are a much more likely route of pervasive exposures to this toxic chemical that are linked to rising allergic sensitivity [3] and tissue injury [4–9]. We found that 10 dermal exposures to 0.5% MI induced pain responses that lasted much longer–up to 70 days after the last challenge. It is likely that the unique tolerogenic stratified epithelium of the vaginal mucosa

necessary to maintain commensal flora [28, 29] led to swifter resolution of allergy-driven pain in our intravaginal study. In contrast, labiar dermal exposures resulted in provoked pain sensitivity that lasted at least three times as long; ongoing studies suggest that sensitivity may last beyond 90 days. It is well-known that the route of exposure greatly affects allergic response outcomes [30]. Given the elevated intensity and prolonged nature of painful responses following dermal exposure versus that of intravaginal challenge that we previously observed, understanding the potentially unique mechanistic distinctions contributing to development of painful responses after MI exposure in these two different tissues could further contextualize the intersections between allergies, mast cells, chronic pain, tolerance, and the risk behind various types of environmental exposures, particularly in the context of vulvodynia.

Mast cell numbers were elevated in the labiar skin after dermal MI exposure. This mast cell accumulation in allergic, painful tissue mirrors findings in vulvar biopsies from some patients with localized, provoked vulvodynia [12]. The small molecule tyrosine kinase inhibitor imatinib depletes mast cells via blockade of the anti-apoptotic c-Kit signals necessary for cell growth and survival *in vitro* and *in vivo* [31–34]; imatinib therapy for cancer is known to cause mast cell deficiencies in treated patients [27]. We found reduction of labiar mast cells as well as tactile sensitivity in MI-challenged mice treated intra-peritoneally with imatinib for six days after 10 MI exposures (Fig 5D). In ongoing studies, we find that preventive administration of imatinib before and during MI exposures also prevents mast cell increases and subsequent painful responses (S2 Fig). Without imatinib treatment, increased mast cell density in the skin lasts up to 60 days after the 10 MI challenges (Fig 3E).

We found increased levels of IL-6 and IL-1β (Fig 1A, 1B, 1D and 1E), and local infiltration of CD4$^+$ regulatory and CD8$^+$ resident memory T-cells in MI challenged skin (Fig 2A–2D). IL-6 and IL-1β are elevated in vestibular tissue from vulvodynia patients [14, 15]. IL-6 has also been implicated in atopic dermatitis and can regulate T cell differentiation [35, 36]. T cells can infiltrate vestibular tissue in vulvodynia patients [13] and central memory T cells are elevated in the blood of persons with MI sensitivity [37], suggesting roles for these immune cells in both types of inflammation. T regulatory cells induce mast cell accumulation in models of allergic asthma [38], and mast cells drive airway remodeling via IFN-γ-dependent pathways [39]. Following MI exposure, IFN-γ transcript levels increased in the skin (Fig 2E) concurrently with CD8$^+$CD103$^+$ memory T cell influx (Fig 2C and 2D). While these T cells do not appear to remain at high numbers in the tissue long term, they likely participate in MI-induced changes that contribute to persistent painful responses. Levels of IgE, a survival factor for mast cells [40], were also elevated in the serum of MI-challenged mice (Fig 3F). Eosinophils were recruited early following MI skin exposure (Fig 1F) and are known regulators of allergic disease and tissue remodeling [41]. We also saw MI-driven accumulation of neutrophils in the skin (Fig 1G); these cells are associated with psoriasis and atopic dermatitis [42, 43]. To our knowledge, eosinophils and neutrophils have not been studied in vulvodynia or in MI contact allergy but may have important modulatory roles in both. We are currently investigating the roles of these and other cells in MI-driven tissue changes.

Understanding the roles of immunomodulation in tissue responses to MI is timely and necessary. MI is an emerging allergen, a strong sensitizer, and ubiquitous in numerous residential and workplace environments. Given the amplified vulvodynia risk caused by allergies and occupational exposure to cleaning solvents, MI may well be one of the missing links in the etiology of this prevalent and debilitating pain condition. While the European Union has already placed restrictions on the use of MI in cosmetics from 100 ppm to 15 ppm [44], use of MI at 100 ppm in personal use products 100 ppm is still permitted in the United States [45]. Our work sets the stage for deeper mechanistic inquiry into the inflammatory response to chronic dermal MI exposure. Such environmental exposures may prove to be important and hitherto

unacknowledged risk factors for chronic allergic inflammation and subsequent chronic pain including, but not limited to, vulvodynia. Understanding the pivotal roles of sentinel mast cells in the tissue changes that result from toxic exposures will aid the development of novel, targeted, effective therapies to treat and manage pain as well as help shape personal choices and broader regulatory policies to minimize and avoid such exposures in the future.

## Supporting information

**S1 Fig. Representative flow plots of gating strategy demonstrate differences in T lymphocyte populations between MI and AOO treated mice.** (A-B) Representative flow plots showing contour plots with 5% outliers T lymphocyte gating strategy in MI-challenged (A) and AOO-treated (B) mice. CD3$^+$ populations were gated out of CD45$^+$CD19$^-$ cells first. CD8$^+$ and CD4$^+$ were gated out of the CD3$^+$ gate. Out of CD4$^+$ cells CD44 and CD25 was evaluated and out of the CD8$^+$ cells CD103 was analyzed. Results were collected on a BD Fortessa and all analysis was done using FlowJo software.
(TIF)

**S2 Fig. Mast cell density and tactile sensitivity decreased after 10 preventative imatinib treatments followed by repeated MI challenges.** For preventative imatinib treatment, mice were treated with 100 μL of 30 mg/kg imatinib dissolved in 0.9% saline injected intraperitoneally 30 minutes before each of 10 daily MI challenges. (A) Density of avidin$^+$ mast cells in 10 μm labiar skin cryo-sections from sensitized mice receiving NT or preventatively treated with imatinib and challenged 10 times daily with MI. Results displayed as mean ± SEM; n = 6/ treatment group. Significance with respect to control group, $^*$ p$<$0.05. (B) Tactile sensitivity in sensitized mice preventatively treated with imatinib or NT and then 10 daily MI challenges. Results reported as mean ± SEM of the percent decrease from baseline in the withdrawal threshold for each treatment group, n = 18/treatment group. Black dotted line denotes 33% decrease from baseline and grey dotted line marks 0% decrease. Significance with respect to control group, $^{***}$ p$<$0.001.
(TIF)

**S1 Table. Average labiar withdrawal thresholds.** Labiar withdrawal thresholds of MI sensitized AOO- and MI-challenged mice (top) or MI-sensitized and challenged NT or therapeutic imatinib treated mice (bottom). Thresholds presented in grams as mean ± SD in mice at baseline prior to sensitization and at predetermined timepoints after challenge (n = 18). Percent change in withdrawal threshold from baseline is depicted in Fig 4 for MI/AOO (10) and MI/ MI (10) mice. Withdrawal thresholds for therapeutic imatinib (MI/MI (10) + Imatinib (6)) or NT (MI/MI (10) + NT) mice are shown in Fig 5.
(DOCX)

## Acknowledgments

The authors thank Jamie Atkins for animal care, Patty Byrne Pfalz for administrative assistance, past and current members of the Chatterjea laboratory for their help and support, and Drs. Bernie Harlow and Vittorio Addona for helpful consultation.

## Author Contributions

**Conceptualization:** Devavani Chatterjea.

**Data curation:** Jaclyn M. Kline, Erica Arriaga-Gomez, Tenzin Yangdon.

**Formal analysis:** Jaclyn M. Kline, Erica Arriaga-Gomez, Tenzin Yangdon, Beebie Boo, Jasmine Landry, Marietta Saldías-Montivero, Nefeli Neamonitaki, Hanna Mengistu, Sayira Silverio, Hayley Zacheis, Dogukan Pasha, Devavani Chatterjea.

**Funding acquisition:** Devavani Chatterjea.

**Investigation:** Jaclyn M. Kline, Erica Arriaga-Gomez, Tenzin Yangdon, Beebie Boo, Jasmine Landry, Marietta Saldías-Montivero, Nefeli Neamonitaki, Hanna Mengistu, Sayira Silverio, Hayley Zacheis, Dogukan Pasha, Tijana Martinov, Devavani Chatterjea.

**Methodology:** Jaclyn M. Kline, Erica Arriaga-Gomez, Jasmine Landry, Tijana Martinov, Devavani Chatterjea.

**Project administration:** Devavani Chatterjea.

**Resources:** Brian T. Fife, Devavani Chatterjea.

**Supervision:** Devavani Chatterjea.

**Validation:** Tenzin Yangdon, Beebie Boo, Jasmine Landry, Marietta Saldías-Montivero, Nefeli Neamonitaki, Hanna Mengistu, Sayira Silverio, Hayley Zacheis, Dogukan Pasha, Devavani Chatterjea.

**Visualization:** Jaclyn M. Kline, Erica Arriaga-Gomez, Tenzin Yangdon.

**Writing – original draft:** Jaclyn M. Kline, Devavani Chatterjea.

**Writing – review & editing:** Jaclyn M. Kline, Tijana Martinov, Devavani Chatterjea.

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
