## [Decision Letter · Decision Letter 0]

14 Aug 2020

PONE-D-20-21835

Repeated dermal application of the common preservative methylisothiazolinone triggers local inflammation, T cell influx, and prolonged mast cell-dependent tactile sensitivity in mice

PLOS ONE

Dear Dr. Chatterjea,

Thank you for submitting your manuscript to PLOS ONE. After careful consideration, we feel that it has merit but does not fully meet PLOS ONE’s publication criteria as it currently stands. Therefore, we invite you to submit a revised version of the manuscript that addresses the points raised during the review process.

We look forward to receiving your revised manuscript.

Kind regards,

Praveen Thumbikat

Academic Editor

PLOS ONE

Additional Editor Comments:

This manuscript by Kline et al., expands on prior work by this group in models of vulvodynia. This particular manuscript follows familiar ground in describing increased perigenital skin sensitivity and a role for mast cells in the methylisothiazolinone induced mouse model. New and noteworthy experiments in this study include the characterization of the immune compartment including T cells, neutrophils and soluble cytokines at various times in the model. In addition, the authors use imatinib as a new agent to inhibit mast cell signaling and show some efficacy in reducing symptoms in the mouse model. While the study itself is well designed and has adequate controls, a number of concerns do remain that would need to be addressed. They are listed below in no particular order.

1. Sections of the introduction are extremely similar/verbatim to a recent publication from the group.

2. In the mechanical hypersensitivity assays shown - all the results are shown after normalization (percent decrease etc). It is difficult to appreciate the relevance of the decreases without seeing the absolute values - perhaps a table can be shown in supplementary data to complement the normalized data shown in figures.

3. The increases in NGF and cadm1 in the spinal cord is interesting, but its direct relevance to the current study is limited and therefore its inclusion in the supplementary data needs to be reconsidered.

Journal Requirements:

2. As part of your revisions please provide additional details pertaining to the care and use of the animals utilized for your research, including: (1) the number of animals as well as sample size justification (power analysis information); (2) details pertaining to monitoring of the animals, humane endpoints and so forth; (3) mortality rate and any unexpected adverse events. Please also complete and submit the ARRIVE Guidelines checklist - Essential 10: https://arriveguidelines.org/resources/author-checklists.

Reviewers' comments:

Reviewer's Responses to Questions

**Comments to the Author**

1. Is the manuscript technically sound, and do the data support the conclusions?

Reviewer #1: Yes

2. Has the statistical analysis been performed appropriately and rigorously? 

Reviewer #1: Yes

3. Have the authors made all data underlying the findings in their manuscript fully available?

Reviewer #1: Yes

4. Is the manuscript presented in an intelligible fashion and written in standard English?

Reviewer #1: Yes

5. Review Comments to the Author

Reviewer #1: This manuscript by Kline J., et al., aims to study the effects of exposure to a common household biocide preservative – methylisothiazolinone (MI) on the labiar skin in the development of genital pain and associated inflammatory changes. In this study, the authors use repeated MI challenges to sensitize the skin and trigger an inflammatory reaction as well as ano-genital sensitivity. They observe increases in inflammatory cytokines, as well as increases in the immune cell infiltrates; including neutrophils, eosinophils, mast cells, and T cells upon MI challenges, which causes a dramatic and persistent increase in ano-genital sensitivity. They conclude the manuscript showing that a therapeutic administration of imatinib causes an alleviation of tactile ano-genital sensitivity, suggesting a possible pivotal role played by mast cells and opens up targets for therapeutic approaches.

The manuscript, as it stands, is however not novel as most of the results and observations reported in the Figures (1, 3, and 4) have been published in their previous publication as cited in the manuscript as Arriaga-Gomez E., et al. [14]. The previously published data showed increase in mast cells, ano-genital sensitivity as well as increased pro-inflammatory cytokines IL6 and IFN-gamma. The only difference between the previous manuscript seems to be the mode of administration of MI, which leads to alteration in the persistence of the ano-genital sensitivity. Furthermore, the previously published article already addressed the issue of the role played by mast cells in driving ano-genital sensitivity using THC as a therapeutic approach.

Major Weaknesses:

1. This study lacks novelty and most of the results have been already previously published by this group.

2. CD44 +CD25+ CD4 T cells as well CD44+ CD8 T cells do not represent T cell subtypes but rather activation states of the T cells. It is important to delineate between T cell subtypes and activation state.

6. PLOS authors have the option to publish the peer review history of their article (what does this mean?). If published, this will include your full peer review and any attached files.

Reviewer #1: No

---

## [Author Response · Author response to Decision Letter 0]

27 Sep 2020

Point 1: This study lacks novelty and most of the results have been already previously published by this group.

We thank the reviewer for their careful and comprehensive reading of the manuscript. We acknowledge the reviewer’s critique disputing the novelty of this study given we have recently published similar findings via intravaginal application of MI. However, we respectfully submit that given the completely different route of MI application, this study 1) validates findings that link occupational exposures to vulvodynia with the skin being a more probable exposure site and 2) demonstrates the capacity of skin responses to MI to be dramatically more pronounced than that within vaginal mucosal tissues. It is well known that route of administration can dramatically affect allergic responses1. However, this has yet to be shown for MI and in particular, to our knowledge it has not been shown to be linked to intensity and duration of MI-induced pain. Furthermore, although there is a growing body of work demonstrating rapid increasing prevalence of MI skin allergy particularly in the US and Europe, published work that characterize immune dynamics of MI skin allergy are minimal. We provide a framework looking at numerous, likely important immune infiltrate at various timepoints during MI allergy which are key first steps to begin to understand immune responses to this preservative, all while linking it to development of painful responses. We have thoroughly revised both our Introduction and Discussion sections to clarify and emphasize these points. 

Point 2: CD44 +CD25+ CD4 T cells as well CD44+ CD8 T cells do not represent T cell subtypes but rather activation states of the T cells. It is important to delineate between T cell subtypes and activation state.

The reviewer raised an excellent point regarding the use of the word subtypes for activated T cells. We thank them for catching this error and have removed “subtypes” and replaced with “activation states.”

---

## [Decision Letter · Decision Letter 1]

12 Oct 2020

Repeated dermal application of the common preservative methylisothiazolinone triggers local inflammation, T cell influx, and prolonged mast cell-dependent tactile sensitivity in mice

PONE-D-20-21835R1

Dear Dr. Chatterjea,

We’re pleased to inform you that your manuscript has been judged scientifically suitable for publication and will be formally accepted for publication once it meets all outstanding technical requirements.

Kind regards,

Praveen Thumbikat

Section Editor

PLOS ONE

Additional Editor Comments (optional):

Reviewers' comments:

Reviewer's Responses to Questions

**Comments to the Author**

1. If the authors have adequately addressed your comments raised in a previous round of review and you feel that this manuscript is now acceptable for publication, you may indicate that here to bypass the “Comments to the Author” section, enter your conflict of interest statement in the “Confidential to Editor” section, and submit your "Accept" recommendation.

Reviewer #1: All comments have been addressed

2. Is the manuscript technically sound, and do the data support the conclusions?

Reviewer #1: Yes

3. Has the statistical analysis been performed appropriately and rigorously? 

Reviewer #1: Yes

4. Have the authors made all data underlying the findings in their manuscript fully available?

Reviewer #1: Yes

5. Is the manuscript presented in an intelligible fashion and written in standard English?

Reviewer #1: Yes

6. Review Comments to the Author

Reviewer #1: The authors have addressed all the critics raised. The manuscript in the current form is well written and concise.

7. PLOS authors have the option to publish the peer review history of their article (what does this mean?). If published, this will include your full peer review and any attached files.

Reviewer #1: No

---

## [Editor Report · Acceptance letter]

16 Oct 2020

PONE-D-20-21835R1 

Repeated dermal application of the common preservative methylisothiazolinone triggers local inflammation, T cell influx, and prolonged mast cell-dependent tactile sensitivity in mice 

Dear Dr. Chatterjea:

I'm pleased to inform you that your manuscript has been deemed suitable for publication in PLOS ONE. Congratulations! Your manuscript is now with our production department. 

Kind regards, 

on behalf of

Dr. Praveen Thumbikat 

Section Editor

PLOS ONE